# A Review of Modern Methods for the Detection of Foodborne Pathogens

**DOI:** 10.3390/microorganisms11051111

**Published:** 2023-04-24

**Authors:** Mohammed Aladhadh

**Affiliations:** Department of Food Science and Human Nutrition, College of Agriculture and Veterinary Medicine, Qassim University, Buraydah 51452, Saudi Arabia; aladhadh@qu.edu.sa

**Keywords:** foodborne, pathogens, bacteria, fungi, viruses, culture-based, PCR, immunoassays, NGS, illnesses

## Abstract

Despite the recent advances in food preservation techniques and food safety, significant disease outbreaks linked to foodborne pathogens such as bacteria, fungi, and viruses still occur worldwide indicating that these pathogens still constitute significant risks to public health. Although extensive reviews of methods for foodborne pathogens detection exist, most are skewed towards bacteria despite the increasing relevance of other pathogens such as viruses. Therefore, this review of foodborne pathogen detection methods is holistic, focusing on pathogenic bacteria, fungi, and viruses. This review has shown that culture-based methods allied with new approaches are beneficial for the detection of foodborne pathogens. The current application of immunoassay methods, especially for bacterial and fungal toxins detection in foods, are reviewed. The use and benefits of nucleic acid-based PCR methods and next-generation sequencing-based methods for bacterial, fungal, and viral pathogens’ detection and their toxins in foods are also reviewed. This review has, therefore, shown that different modern methods exist for the detection of current and emerging foodborne bacterial, fungal, and viral pathogens. It provides further evidence that the full utilization of these tools can lead to early detection and control of foodborne diseases, enhancing public health and reducing the frequency of disease outbreaks.

## 1. Introduction

Microorganisms are important in food production, safety and spoilage. Microbial interactions with food can lead to food unfit for human consumption [1,2]. This is because some of these foodborne microorganisms not only cause food spoilage but are also pathogenic in nature, presenting risks to food safety and foodborne illnesses to food handlers, consumers, and society in general [3].

Food-borne illnesses and diseases are major threats to human health and lives with over 200 foodborne diseases identified [4]. Common foodborne pathogens include bacteria such as *Bacillus cereus*, *Clostridium botulinum*, *Campylobacter* sp., *C. perfringens*, some *Escherichia coli serogroups*, *Listeria monocytogenes*, *Salmonella* spp., *Shigella* spp., *Staphylococcus aureus Vibrio* spp. etc. [5,6,7]. Consequently, detection methods were developed for these pathogens to ensure compliance with fixed legal and regulatory food safety thresholds based on either pathogen presence or absence and counts. Foodborne fungal species such as *Penicillium*, *Claviceps*, *Aspergillus* and *Fusarium* species which can produce mycotoxins are also a threat to public health [6]. Key foodborne pathogenic viruses include Norovirus, Hepatitis A and E viruses, Rotavirus, Enterovirus, Adenovirus, Sapovirus, Achivirus, and Astrovirus [7,8].

In recent times, there have been significant advances in food preservation methods designed to extend the shelf-life of food and eliminate spoilage and or pathogenic microbial groups. Some of the commonly used methods include storage at low temperatures (chilling and freezing), application of chemical preservatives (e.g., essential oils and bacteriocins [9,10,11,12], and vacuum and modified atmosphere packaging [13,14]. While some of these methods eliminate food pathogens, research has shown that when some of these pathogens are exposed to environmental stress or food preservation-related stress, they may survive the stress and exist as metabolically injured cells with impaired metabolic activities. In this case, the pathogens may or may not recover from this damage (irreversible or reversible). They may also survive as dormant cells, for example as viable but not culturable cells (VNBC) or as persister cells [15]. VNBC cells are cells that are no longer culturable using known growth media but possess low and detectable metabolic activities while persister cells are surviving cells tolerant to but not resistant to antibiotics, with low metabolic activities, undetectable with viability assays. However, persister cells may become culturable when exposed to a specific stimulus [15].

The regular occurrence of foodborne diseases shows that despite the significant advances in food manufacturing and safety, foodborne pathogens still constitute significant public health risks [15]. Consequently, a variety of methods, culture, and culture-independent, have been developed to detect these foodborne pathogens in food and protect public health. There have been numerous reviews of foodborne pathogens’ detection using culture-dependent and independent methods such as the use of culture media, Polymerase Chain Reactions (PCR) (multiplex PCR, Real-Time PCR, Reverse Transcriptase PCR (RT-PCR)), DNA microarray, nucleic acid sequence-based amplification (NASBA), and Next Generation Sequencing (NGS) immunological and nanotechnology-based methods [15,16,17].

However, most of these reviews are skewed towards bacteria (with a passing mention of other foodborne pathogenic groups). This is because bacteria are the most encountered foodborne pathogens of public health significance. However, other pathogenic groups such as fungi and viruses are also important as they also are known to cause foodborne diseases outbreak. Therefore, this review not only focuses on bacteria but also equally on fungi and viruses with emphasis on the use of some specific molecular methods such as PCR (End-point PCR, Multiplex PCR and RT PCR etc.) and NGS to advance our current knowledge of the detection and identification methods of these different foodborne pathogens. The latest data on the occurrence of foodborne diseases are reported with some examples of recent outbreaks of foodborne diseases being presented in the next section to demonstrate that foodborne pathogens continue to pose a significant risk to public health.

## 2. Outbreaks of Foodborne Illnesses

### 2.1. Bacteria

Foodborne illnesses caused by bacteria are a worldwide occurrence. For example, between 128,000 and 325,000 hospitalizations and 3000 to 5000 deaths, and at least 76 million illnesses per year are estimated to be associated with foodborne illnesses in the USA, costing the economy up to $83 billion per year [4,18,19]. In the EU, specifically in 2020, there were 20,017 human cases and 3086 foodborne outbreaks of foodborne diseases, with *Salmonella* (in eggs and egg products) being the most frequently implicated bacterial causative agent for foodborne outbreaks [20]. Apart from *Salmonella,* and *Campylobacter*, *Yersinia*, Shiga toxin-producing *Escherichia coli* (STEC) and *Listeria monocytogenes* were the other most common causative agents of foodborne illnesses in 2020 in the EU [20]. *L. monocytogenes*-related illnesses were one of the illnesses that resulted in the highest human fatalities in the EU during this time frame [20]. Worldwide, it has been estimated that there are about 3 million cases of diarrhoea related to foodborne microbial pathogens annually [19]. According to the Australian Department of Health, it is estimated that 4.1 million foodborne diseases or cases occur in Australia [21]. A review of foodborne pathogens and associated illnesses have been drawn up by authors [5].

### 2.2. Fungi

In contrast to bacteria and viruses, outbreaks of illnesses associated with fungal foodborne pathogens are rare given that only about 300 of the estimated 1.5 million fungal species are known to cause illnesses in humans [22]. Nevertheless, there are some notable examples of the outbreak of illnesses, typically due to fungal secondary metabolites such as toxins and sometimes occurring only in vulnerable populations such as people with suppressed immunity, with transplants or undergoing immunosuppressive treatments. However, on rare occasions, some immunocompetent people can also succumb to foodborne fungal illnesses.

For example, in 2013, there was an outbreak of gastroenteritis (vomiting, nausea and diarrhoea) in over 200 people who consumed yoghurts contaminated with *Mucor circinelloides* in the USA [23]. An outbreak of food poisoning by *Rhizopus microsporus* was reported in 7 hospital patients in Hong Kong from contaminated pre-packaged ready-to-eat meals or cornstarch used to produce allopurinol tablets [24]. A review of the literature [25] has shown some key filamentous fungi such as *Aspergillus*, *Fusarium*, and *Mucor* when consumed with contaminated food or inhaled can cause localized infections in the sinuses, lungs, and other areas in immunocompetent people. Invasive pulmonary disease caused by some of these fungal pathogens typically occurs in immunocompromised patients (with inhalation as a primary source). However, gastrointestinal routes could be important as well, hence the need for such patients to avoid foods likely to be contaminated with fungi [25]. An extensive survey of foodborne fungal agents was drawn up by authors [26].

### 2.3. Viruses

Viral outbreaks of food poisoning are less frequently reported compared to bacterial outbreaks. Concerning viruses, Noroviruses and Hepatitis A viruses are one of the most common foodborne viral pathogens known to cause illnesses in humans. Noroviruses, in particular, have been linked to up to 21 million cases of acute gastroenteritis annually in the United States of America [27]. In 2020 in the EU, Norovirus (in crustaceans, shellfish, molluscs, and their products) was a major cause of reported food outbreaks [20]. Enteric viruses are thought to account for up to 13% and 45% of outbreaks of foodborne illnesses in the EU and the US respectively [28] with sporadic reports of these outbreaks in the public domain. A list of foodborne illnesses associated with viruses has been reported by [8]. 

A recent review of reported virus-linked foodborne illness outbreaks has identified that the gastroenteritis outbreak at the 2018 Winter Olympics in South Korea (194 cases), the consumption of frozen raspberry-linked outbreak in 2017 in Canada (700 cases) and the outbreak in 2017 Royal Caribbean cruise line were all linked to Noroviruses [27]. Other reviews of the literature have shown that in 2014, viral agents accounted for 20% of all reported outbreaks in the EU [20]. Specific examples include the 162 Hepatitis A virus cases in the US in 2011 (consumption of contaminated pomegranate seeds) and over 1100 cases in China in 2012 (Norovirus in frozen strawberries). In 2020, West Nile virus-based illnesses were one the major foodborne pathogens that resulted in the highest number of human fatalities in the EU [20].

## 3. Methods for Detecting Foodborne Pathogens

The detection of pathogens in food matrices with high reproducibility and sensitivity is of utmost importance to guarantee food safety. However, it is still a difficult challenge due to factors such as interference by other and non-target microbiota, low numbers of target groups, and difficulties in microbial extraction from food matrices. There are different methods used for detecting and identifying foodborne pathogens. These range from the use of culture-based methods, immunological assays, nucleic acid-based methods (Polymerase Chain Reactions (PCR) and Next Generation Sequencing (NGS) methods. This review provides a brief explanation of the principles underpinning each class of method (excluding biosensors), where relevant, followed by a review of the application of some of the frequently used approaches for pathogen detection in the current literature. This paper reviews the use of culture-based methods, immunological assays, PCR and Next Generation Sequencing-based approaches. 

However, the success of any of these approaches is dependent on the use of the appropriate aseptic sampling and sample storage protocols. The sampling method is dependent on the type of food being sampled, target microbial groups, and the microbial detection methods. For example, sample collection and analysis for any method involving the isolation and use of microbial cultures should follow standard protocols certified and developed by official organizations such as FDA, FSIS/USDA, ISO, and AOAC [29]. For other methods, standardized protocols exist for sample collection, analysis, and data interpretation based on the kit, reagent, and equipment manufacturer’s prescription.

### 3.1. Culture-Based Methods

Culture-based methods remain the reference methods for detecting foodborne pathogens despite the existence of other more modern methods [30,31,32,33]. These methods are premised on the ability of bacteria and fungi to grow on culture media, forming visible colonies which can then be subject to other downstream assays as required. They remain the first choice and may be required by law for the detection of microorganisms in food testing laboratories [15]. They can be used to generate culture-based qualitative and quantitative data on the presence of food pathogens. Culture-based methods are more successful when the growth requirements of target microorganisms are known with culture media being used to enrich, selectively isolate, or discriminate between target microorganisms and other groups [6]. In addition, cultures can be subject to tests such as colony characteristics, Gram staining reaction, biochemical characterization, and MALDI TOF MS and PCR-sequencing for identification purposes [6].

Despite its ease of use, relatively cheap cost, and the fact that it allows the isolation of the microorganisms for use in downstream applications, culture-based methods have significant limitations (Table 1). These methods have low sensitivity as not all microorganisms are culturable and some culturable bacteria (such as *S. typhi* and *E. coli*) can exist as VBNC, as a result of stress and other factors [15,34]. These can lead to underestimation or non-detection of foodborne pathogens creating a potential risk to food safety. Culturing microorganisms is also a slow and laborious process requiring a series of steps and may require the use of adjunct methods (e.g. biochemical, serological, nucleic acid-based methods for conclusive identification) and can take up to a week for bacteria [16,35] or longer for fungi. They also may not be suitable for the rapid detection of microorganisms for on-the-spot or real-time foodborne pathogen detection [36,37].

#### 3.1.1. Culture-Based Detection of Foodborne Bacteria

Some of the key bacterial groups commonly associated with foodborne illnesses are *E. coli*, *L. monocytogenes*, *Yersinia*, *Salmonella*, *Campylobacter*, *Clostridium*, *Enterobacteria*, and *Bacillus species*. These bacterial groups cause foodborne illnesses/diseases by direct bacterial growth and or the secretion of toxins. These bacterial groups can be cultured using the appropriate culture media except when they are in the VBNC phase. They are usually ingested through the consumption of contaminated foods such as dairy, poultry, and beef products (including Ready to Eat (RTE) foods) [7]. Therefore, the use of cultural media alone is unreliable. Moreover, they are laborious and have limited sensitivity as they can only be used to detect a subset (culturable bacteria) with a significant population of bacteria not being culturable. Indeed, nucleic acid-based and other molecular methods with greater sensitivity and ability to detect both culturable and non-culturable bacteria should be more widely used. Instead, culture-based approaches coupled with other methods such as PCR, Immunoassays, NGS, Biosensors, and MALDI Tof MS are increasingly being used for the detection and identification of foodborne pathogens. This reduces the time needed for bacterial detection/identification from a week using traditional approaches to less than 3 days. For example, MALDI TOF MS is an accurate, sensitive, and relatively faster identification method for fastidious bacteria than PCR and NGS-based approaches [41,42]. It is also a comparatively cheaper method to use compared to NGS [43]. It involves the generation of the peptide mass fingerprint (PMF) of isolated bacterial cultures (and comparing that with the PMF in a selected database for identity typing [43].

In recent times MALDI TOF MS has been successfully used to identify foodborne pathogenic bacteria, especially those belonging to the key foodborne bacterial pathogenic groups. With respect to *Salmonella*, *Campylobacter*, and Enterobacteria, *S. typhimurium* from minced beef [44], *Campylobacter jejuni*, *E. coli*, and *Enterobacter cloacae* from Matazeez, a Middle Eastern lamb and vegetable-based stew [45] have been identified using MALDI TOF MS. MALDI TOF MS has also been used to identify *E. coli*, *S. typhimurium* , *Klebsiella*, *Enterobacter* and other food pathogens from fish, meat and dairy products from stores and supermarkets in Egypt [46]. Similarly, MALDI TOF was used to identify foodborne pathogens such as *E. coli*, *S. typhimurium* , *S. aureus*, *C. jejuni*, *Klebsiella*, *and L. monocytogenes* in fresh vegetables in Beijing, China [47], and salads, burgers, and tortillas in Saudi Arabia [45] and raw milk from public markets in Turkey [48]. With respect to *Clostridium*, *C. difficile* and *C. perfringens* from baby food and baby food supplements [49] have been identified with MALDI TOF MS, These reports, therefore, show that despite its limitations culture-based methods coupled with MALDI TOF MS-based approaches are currently being used for the isolation and subsequent identification by MALDI TOF MS of foodborne pathogenic bacteria.

#### 3.1.2. Culture-Based Detection of Foodborne Fungi

Fungi and their metabolites (mycotoxin) can be ingested with contaminated food. In culture-dependent detection, fungi may take up to 7 days of incubation to be visible on culture plates. Moreover, for fungal pathogens, culture-based approaches are now widely used in conjunction with methods such as PCR, Immunoassays, and MALDI TOF MS, NGS for fungal (including pathogenic fungi) detection and identification [50,51,52].

Using MALDI TOF MS as an example, the method has been used to identify detected key fungal groups such as *Rhizopus*, *Aspergillus*, *Fusarium*, and *Mucor* [45]. Potential food pathogens such as *A. niger*, *A. flavus*, and other fungal groups such as *Alternaria alternata*, *P. digitatum*, and *Candida albicans* have been detected and identified in foods such as salads, burgers, tortilla, cheese, *kabsa*, and *jareesh* from restaurants in Al-Qassim regions in Saudi Arabia using MALDI TOF MS [45]. Other fungal groups such as *Mucor* [50], foodborne yeasts [53], and fungi in fermented foods [54] have also been detected and identified using MALDI TOF MS. However, this approach is comparatively less used for fungi than for bacteria primarily because of the limited availability of curated fungal spectra database and adaptation of the method for fungi [55].

#### 3.1.3. Culture-Based Detection of Foodborne Viruses

Noroviruses and Hepatitis A viruses are the most common cause of foodborne illnesses/diseases [20]. Noroviruses alongside other viruses such as Rotaviruses and Astroviruses typically cause gastroenteritis while foodborne Hepatitis viruses cause either hepatitis A or E with the enteroviruses causing hand and foot diseases, meningitis/encephalitis, and heart disorders. They can contaminate food such as seafood (shellfish, oysters, and molluscs), fish, milk, fruits, and vegetables with transmission through the faecal-oral route or inhaled as droplets or through contact with contaminated items [8,56]. The only exception is the Hepatitis E virus which is associated with raw and undercooked meat and liver [8]. Detailed information on the different foodborne viruses, their route of entry, and the type of food they have been detected in have been reviewed [8].

Detection of viruses in food can be carried out by propagating the extracts from test food samples in cell cultures [27] and the observation of the formation of viral-induced cytopathic effects. Viral quantification can then be achieved through tissue culture, infectious dose 50 (TCID 50), plaque assay, and the most probable number method [57]. These methods have been applied in multiple studies to detect viruses of significant public health risks in food items. However, the use of these culture-based methods for the detection of viruses is limited by a variety of factors. These factors include the typically low concentration of viruses in food, thereby requiring the use of very sensitive methods for enrichment before detection [58]. It also takes days to culture viruses, making this approach unsuitable for rapid testing. In addition, some viruses do not show cytopathic effects and efficient cell culture systems do not exist for some of the foodborne viruses such as Noroviruses [27,57,59]. This is why molecular methods such as RT-PCR which are comparably less laborious, highly sensitive, and accurate are more widely used for viral detection. However, there are still some reports of culture-based approaches still being used for the detection of some viruses.

For example, Avian Influenza viruses (H7N9 and H5N8), a substantial risk to public health have been detected in wild birds, poultry, and ducks [60,61,62]. The method of detection typically involves homogenizing samples and growing aliquots of the homogenate in embryonated chicken eggs (cell culture) for 2–5 days followed by hemagglutination, with positive hemagglutination demonstrating the presence of the virus [63]. Sometimes, culture methods are integrated with other tools such as plaque assay methods and integrated cell culture RT-qPCR (virus propagated in cell culture and its amplification assessed with RT-qPCR) to aid viral detection. A good example is the Human adenovirus (HAdv) which has been successfully detected in fresh produce such as lettuce onions and strawberries using plaque assay and integrated cell culture RT-qPCR approaches [64].

### 3.2. Immunological Assays

These assays include serotyping, immunofluorescence, use of lateral flow devices (LFD) and enzyme-linked immune sorbent assay (ELISA) approaches, with ELISA, being a very accurate immunological method for detecting foodborne pathogens and their toxins. These approaches work on the principle that there is an affinity between microbial antigens and antibodies and that this affinity can be exploited for the rapid and accurate detection of foodborne pathogens (Table 1) [37]. The key advantages of these assays are that they are easier to carry out, faster than culture-based methods, can detect toxins, and can be highly specific. However, the contamination of the reaction matrix can lead to the generation of false positives (Table 1) [65]. ELISA and lateral flow devices (LFD) are some of the most widely used immunoassays for detecting foodborne microorganisms and their toxins in recent times with their principle and mode of action well reviewed [16,65,66] ELISA can sometimes be used in conjunction with other methods (including PCR), to increase their specificity and efficiency. For example, a PCR-ELISA approach has led to a 100-fold increase in the detection of *Fusarium verticillioides* in contaminated corn samples compared to detection using conventional PCR [67].

#### 3.2.1. Detection of Foodborne Bacteria and Bacterial Toxins Using Immunoassays

ELISA has been used to detect key bacterial groups such as *Salmonella*, *Campylobacter, E. coli,* and *Listeria*. In particular, ELISA had been used to detect *Salmonella* sp. in meat and dairy products, pasta and eggs [68,69], *Campylobacter* in food samples [70] *E. coli* O157: H7 in cabbage [71] and beef [72], *Vibrio parahaemolyticus* in seafood [73] and *Listeria* in milk [74]. The method has also been used to detect staphylococcal enterotoxins [75], botulinum toxins in spiked food (meat and milk) [76], Shiga toxins [77], and *B. cereus* enterotoxins [78] in different food matrixes.

#### 3.2.2. Detection of Foodborne Fungi and Fungal Toxins Using Immunoassays

ELISA kits (PCR-ELISA) have been developed for the detection of pathogenic *F. verticillioides* [67] and other species. In addition, it is commonly used to detect mycotoxins. ELISA has been used to detect aflatoxin B1 from *A. flavus* in stored maize [79], low levels of aflatoxin in dried stockfish [80], peanuts [81], and soy milk [82]. Similarly, LFDs have been used to detect mycotoxins such as aflatoxin, deoxynivalenol, and fumonisins in maize and barley [66].

#### 3.2.3. Detection of Foodborne Viruses Using Immunoassays

Immunoassays are more commonly used to detect pathogenic viruses in clinical and environmental samples for point-of-care and disease diagnostic and public health surveillance reasons [38,83] than those from food sources. Historically, ELISA immunoassay kits are available for the detection of adenoviruses, group A Rotaviruses, and Astroviruses [84] and have been used for detecting viruses in food items [85]. However, they are no longer a method of choice because of the advent of nucleic acid-based methods which are more rapid and sensitive approaches.

### 3.3. Nucleic-Acid Based Methods (Polymerase Chain Reaction (PCR) and Its Variants (PCR, RT-PCR etc.)

PCR and its variants are nucleic acid-based methods that are used to detect specific DNA or RNA sequences of pathogenic microorganisms. Specific primers are designed for target pathogens allowing them to be exponentially amplified in food samples. PCR-based methods are faster, more sensitive, highly reproducible, and versatile approaches compared to most culture-based and immunoassay methods [65] making it a method of choice for the detection of foodborne pathogens. They are applied to nucleic acids extracted from food samples or microbial isolates isolated from food samples. The principles, advantages, and limitations of PCR-based approaches and their variants are well-reviewed (Table 1) [16,65,86,87].

There are many variants of this nucleic acid-based approach (Figure 1). These include the conventional PCRs (nested, touchdown, hot-start PCR, etc.), which are commonly applied for the detection of foodborne pathogens based on the use of primers targeting the DNA of these pathogens. Nested PCR is a modification of standard PCR and involves the use of 2 different primer sets in two PCR reaction runs. The second primer set is used to amplify a secondary target with the amplicon generated with the first primer set [86]. Nested PCR is, therefore, designed to pick up low levels of target pathogens [88]. Touchdown PCR prevents the amplification of non-specific sequences. This is achieved by starting the early PCR steps at high annealing temperatures followed by incremental decreases in subsequent cycles thereby allowing the primer in use to bind to the target sequence at the highest temperature that prevents the amplification of non-specific sequences [86]. Another variant, multiplex PCR can be used to detect multiple pathogens at the same time as it involves the use of different primers in the same reaction [86]. Reverse Transcriptase PCR (RT-PCR) allows for the detection of RNA (metabolically active microorganisms) by creating a complementary DNA (cDNA) from the RNA transcript after which the cDNA is amplified. Real-time or quantitative PCR (qPCR), is used to quantify pathogen load on food items based on the use of fluorescent dyes or probes [88] using a specially designed thermal cycler. It differs from conventional PCR in that amplification of target DNA is monitored in real-time instead of at the endpoint as obtained in conventional PCR. RT-PCR and qPCR can be combined (RT-qPCR) for qualitative and quantitative purposes. RNA is reverse transcribed into cDNA and the cDNA is subject to Real-Time PCR for quantitative detection of RNA [86]. Other forms of PCR such as droplet digital PCR (ddPCR) allow for quantification without the use of a standard curve [89,90]. In DDPCR, target samples are fractionated into thousands of droplets (each droplet being one-nanolitre reverse micelles (water in oil)) which are subject to a fluorescent probe-based PCR assay [89,90]. Loop-mediated isothermal amplification (LAMP) is another DNA amplification method that yields more DNA copies (about a billion copies) than normal PCR ( about a million copies) within an hour [91]. It involves the use of up to three primer pairs to target up to eight specific sites on DNA strands of interest in amplification reactions conducted at stable temperatures [91].

#### 3.3.1. Detection of Foodborne Bacteria Using PCR-Based Methods

Conventional PCR and its variants (which are DNA based) involve the use of primers (specific or degenerate) that can be applied to detect a target or group of microorganisms in food items. However, these approaches (endpoint PCR, nested PCR, multiplex PCR, etc.) cannot indicate whether the target organism(s) is viable or not. These approaches have been used to successfully detect key bacterial pathogens such as *Salmonella* spp, *Campylobacter* sp., *E. coli* O157:H7, *S. aureus*, *B. cereus*, *L. monocytogenes*, and *V. parahaemolyticus* in ready-to-eat Korean foods [93], *Minas Frescal* cheese [94], milk [95], beef [96], spiked chicken egg, pork, salad [97], and fish [98].

Real-Time or quantitative PCR is more widely used than (PCR or multiplex PCR) to detect and quantify the abundance of key foodborne bacterial pathogens in extracts from food samples or purified isolates from food samples. *Salmonella*, *E. coli*, *Campylobacter*, and *L. monocytogenes* in cheese, chicken, beef burgers, turkey, pork, egg, chicken, and fish have been detected and quantified using Real-Time PCR [34,99,100,101]. Using the same method, *C. perfringens*, *E. coli*, and *S. aureus* were detected and quantified in fresh and ready-to-eat vegetables [102]. Other pathogens such as *V. vulnificus*, *V. parahaemolyticus*, and *V. cholerae* [103,104,105] have been detected in seafood, shrimp, and mussels using Real-Time PCR.

In contrast, Reverse Transcriptase PCR (RT-PCR)-based methods are not as widely used for detecting foodborne bacterial pathogens as Real Time or quantitative PCR. This is because, mRNA despite being a better indicator of the viability of bacteria, is rapidly degraded in the food matrix generating false negative results [106]. The process is also labour-intensive [106] and both of these factors account for why the approach is less used for detecting foodborne bacteria than other methods [15]. Nevertheless, this approach has been used to detect bacterial pathogens in different food samples such as *S. typhimurium* in artificially spiked jalapeno and serrano peppers [107] and *S. enterica* in spiked egg broth and milk [108]. 

Viable PCR (vPCR), another variant of PCR that involves the use of intercalating dyes such as ethidium monoazide (EMA) and propidium monoazide (PMA) for sample pre-treatment followed by PCR amplification allowing for the detection of living foodborne pathogens [109]. vPCR could be more readily applied for the detection of pathogens than RT PCR. These intercalating dyes can penetrate the membrane of damaged or dead bacterial cells, irreversibly binding to their DNA molecules thereby preventing their amplification by PCR primers. Therefore, any amplicon observed from the subsequent PCR will be from live bacterial cells [109]. It has been used to detect live *Salmonella* sp. in spiked pork meat and RTE salad [110] and *Helicobacter pylori* in retail pork meat [111] demonstrating the usefulness of this approach in assessing food safety [110]. Combining vPCR with Real-Time PCR has allowed for the accurate detection and quantification of viable *Campylobacter* sp. in frozen and chilled poultry meat [112] and *Listeria* sp. in spiked meat, salad, cheese, and milk [113]. Other forms of nucleic acid-based amplification methods such as LAMP (Loop-Mediated Isothermal Amplification) have been used to detect *V. vulnificus* (Yang et al., 2021), *V. parahaemolyticus* [103] in foods such as seafood.

#### 3.3.2. Detection of Foodborne Fungi Using PCR-Based Methods

A review of the literature showed limited reports (compared to bacteria) of the detection of pathogenic fungi on food materials using PCR-based methods. However, there are multiple reports of the detection of fungal pathogens from clinical and environmental samples using nucleic acid and PCR-based methods [114,115,116,117]. Different PCR or nucleic acid-based methods such as endpoint PCR, real-time PCR, nested PCR, quantitative (RT)-PCR, loop-mediated isothermal amplification (LAMP) and multiplex PCR have been applied for the detection of pathogenic or mycotoxigenic fungi [118]. These methods have been applied directly on extracts from food samples, swabs of food surfaces, and isolates from food samples to detect and characterize foodborne fungal pathogens. 

Multiplex PCR has been developed and successfully used to detect aflatoxigenic *Aspergillus* isolates from *“meju”*, a traditional fermented soybean food starter from Korea [119]. In this instance, specially designed primers were used to successfully discriminate between aflatoxin and non-aflatoxin-producing fungi, with this being validated by TLC and HPLC results of filtrates from test cultures [119]. Multiplex PCR has also been used to simultaneously detect *Aspergillus*, *Penicillium*, and *Fusarium* in contaminated maize grain powder [120].

Patulin (toxin) producing Penicillium expansum has also been successfully detected on artificially contaminated apples using RT-PCR [121]. Loop-mediated isothermal amplification (LAMP) assays have been developed for the rapid detection of ochratoxin-producing *Penicillium nordicum* in dry-cured meat products [122]. PCR combined with other methods such as denaturing gradient gel electrophoresis (DGGE) and sequencing can now be used to detect ochratoxin-producing Aspergillus niger in wine [123]. DGGE is a method that is used to separate DNA fragments (PCR amplicons) based on their melting characteristics in polyacrylamide gels. This generates a gel-based fingerprint of the key microbial groups amplified and further downstream processes can be used to characterize the identity of these groups if required [123].

#### 3.3.3. Detection of Foodborne Viruses Using PCR-Based Methods

Molecular methods such as PCR, Multiplex PCR, Real Time-PCR (RT PCR), digital RT-PCR, and Quantitative RT PCR can also be used to detect viral pathogens in food. RT-PCR, especially, is a commonly used method probably because a significant number of viruses are RNA viruses and one of the best approaches for the detection of foodborne viral pathogens [57]. This is because it allows for the quantification of viral particles when combined with quantitative real-time PCR, which is very sensitive and specific with high throughput [27]. These can be applied directly to food samples or coupled with cell culture assay (isolation of pathogens using cell culture followed by detection using PCR) [57]. Either way, viruses must be extracted and concentrated from food materials first, before the application of PCR. The limitations of the technique are that the extraction processes may be inefficient resulting in low recovery or outright loss of viral particles [27] and the amplification process may also be affected by sample inhibitors. RT PCR or real-time quantitative PCR (RT qPCR) is also unable to distinguish between infectious and non-infectious particles [59]. One way to overcome the limitations of the detection concerning identifying infectious particles is the use of intercalating dyes such as propidium monoazide (PMA) or ethidium monoazide (EMA) for sample pre-treatment before RT PCR or RT qPCR to inhibit the amplification from non-infectious particles. This has been successfully used to distinguish infectious Hepatitis A viruses and Rotavirus from non-infectious particles in laboratory-based analysis [124].

There are many examples of the use of RT-PCR for viral detection in food. For example, the prevalence of Hepatitis A and Norovirus was investigated in harvested mussels in Italy. RT-PCR was used to show the high prevalence of norovirus conclusively demonstrating the public health risk associated with mussel consumption [125]. RT-PCR has similarly been used to detect zoonotic Hepatitis E viruses in spiked pork liver sausages [126] and in raw and liver sausages sold in retail shops in Germany [127]. In Brazil, this approach was used to assess the prevalence of Adenovirus, Hepatitis E virus and Rotavirus in beef, pork, and chicken meat cuts from a market in Brazil with results showing that Rotavirus was the most significant viral pathogen in chicken cuts in these samples [128].

RT-PCR has also been used along with new methods such as microfluidics to detect viruses in different food items or used to enhance the viral detection efficiency of other molecular methods. Raw fruits such as soft berries can harbor pathogens such as Norovirus and Hepatitis A virus which are causative agents of gastroenteritis. However, their concentrations and infectious doses tend to be low in contaminated food samples. Therefore, microfluidics was successfully applied to enhance the detection of these two viruses in soft berries using digital RT-PCR and in this instance, it lowered PCR inhibition while boosting viral detection efficiencies [129]. Similarly, microfluidic digital PCR was successfully used to detect Norovirus and Hepatitis A viruses in spiked/contaminated lettuce, with the observed viral recoveries significantly higher than those obtained with other molecular methods such as RT-PCR [130].

### 3.4. Next-Generation Sequencing (NGS) Methods

Next-generation sequencing (NGS) approaches coupled with bioinformatics are powerful approaches that have greatly benefitted food safety. NGS is primarily used in many ways, firstly to determine the whole genome sequence of an isolate (whole genome sequencing or WGS) and secondly, in metagenomics to determine the sequences of many of the microorganisms present in a sample. In the latter application, particular microbial groups (bacteria, fungi, or viruses) can be targeted using 16S rRNA, ITS, or any other biological markers [131]. The different workflows of NGS applied to food microbiology are shown in Figure 2. There are excellent reviews of Next Generation Sequencing methods principles, types, advantages, and disadvantages (Table 1) [39,40,132].

#### 3.4.1. Detection and Identification of Foodborne Bacteria Using NGS

NGS has many applications in food safety. NGS has been applied to successfully screen fresh produce for the presence of human pathogens such as *Salmonella* a, allowing for their detection [134]. The limit of detection (LoD) of *Salmonella* and MS2, a Norovirus surrogate was determined using multiple NGS approaches (16S amplicon sequencing, RNA-seq using ScriptSeq, and NEBNext (New England BioLabs ) kits. ScriptSeq approach was the most sensitive method with a *Salmonella* LoD of 10^4^ CFU reaction^−1^ (Colony Forming Units) and the phage MS2 LoD of 10^5^ PFU reaction^−1^ (Plaque Forming Units) [134]. A review of the literature has shown that *L. monocytogenes strains* have been identified in foodborne disease outbreaks using WGS [135]. Shiga toxin-producing *E. coli* have been detected in spiked spinach using shotgun metagenomics with an LoD of ~10 CFU/100 g) [136].

NGS has also been applied after the detection of foodborne bacterial pathogens by other methods such as RT PCR, for confirmatory purposes. Pathogens such as *S. sonnei*, *L. monocytogenes*, *C. jejuni*, *S. enterica* subsp. enterica serovar *enteritidis*, *S. aureus*, *E. coli*, and *Y. enterocolitica* have been detected using multiplex PCR-based NGS (Illumina sequencing) from food items such as meat (pork and chicken), milk, seafood, and vegetables [137]. Similarly, NGS (amplicon-based Illumina sequencing) has been successfully applied to amplicons from PCR and RT-PCR of samples from RTE salads, detecting opportunistic pathogens such as *Aeromonas hydrophyla* and *Rahnella aquatilis* [138]. NGS in form of whole genome sequencing is especially useful for the surveillance of foodborne pathogens [139].

#### 3.4.2. Detection of Foodborne Fungi Using NGS

NGS is comparable to less applied to fungal pathogens than bacterial pathogens. However, when applied, it has been used for the detection and identification of pathogens from clinical samples and, to some extent, environmental samples [140,141].

#### 3.4.3. Detection of Foodborne Viruses Using NGS

Presently, NGS methods are used as part of a suite of methods for the detection and sequencing of foodborne viruses. This means that non-NGS methods such as RT PCR are first used to detect positive samples before these samples are sequenced with NGS for confirmatory purposes. For example, whole genome sequencing was used to phylogenetically characterize the Avian influenza virus obtained from the chicken and duck meat products brought in by international travelers [63] and samples from Tongzhou poultry meat markets (China) [142]. NGS (metagenomics) was also applied to frozen strawberries in a Norovirus outbreak in Germany in 2012 (alongside other methods such as RT qPCR and dPCR) to conclusively detect and type the genotype of the Norovirus involved in the outbreak [143]. Similarly, NGS (metagenomics) was used to successfully detect Norovirus and or Hepatitis A viruses in RT-PCR-positive samples from virus-spiked celery samples [144], oysters from fish producers in Japan [145] and in RT qPCR-positive lettuce, parsley, and strawberry samples from organic farms in Spain [146]. There have also been instances where the amplicon-based NGS approach has been used to detect and type viral pathogens in food samples. For example, a high diversity of Noroviruses and Hepatitis A viruses was detected in organically grown fresh lettuce, parsley, and strawberry in RT qPCR positive samples using nested PCR targeting NoV and HPV genes with Illumina adapters incorporated into the nested primers [146].

## 4. Conclusions and Future Directions

A review of four approaches, culture-based, immunoassay, nucleic-acid-based (PCR), and NGS-based methodologies was conducted. Here, culture-based methods when allied with other techniques such as MALDI TOF MS, and nucleic acid-based methods such as PCR (Real-Time PCR for Bacteria and RT-PCR for fungi) are effective at rapidly detecting and identifying different foodborne pathogens, especially those belonging to bacterial and fungal groups. Nucleic acid-based methods such as Real-Time PCR and vPCR combined with sequencing approaches are more widely used than immunoassay and NGS-based approaches for pathogen detection. NGS-based approaches (such as metagenomics) offer unparalleled insights into the genotype, diversity, and activities of foodborne pathogens with their potential for pathogen surveillance, tracing, screening (in a food chain), and identification. These approaches would be extremely useful in the earlier detection, and management of foodborne outbreaks thereby enhancing public safety and health. Current and future directions should be focused on ensuring that existing tools are applied for foodborne pathogen control through training of food surveillance and safety officers on the use of these methods. In addition, more research should be carried out on the development of more effective approaches (such as the use of bacteriophages for the control and elimination of foodborne pathogens.

## Figures and Tables

**Figure 1 microorganisms-11-01111-f001:**
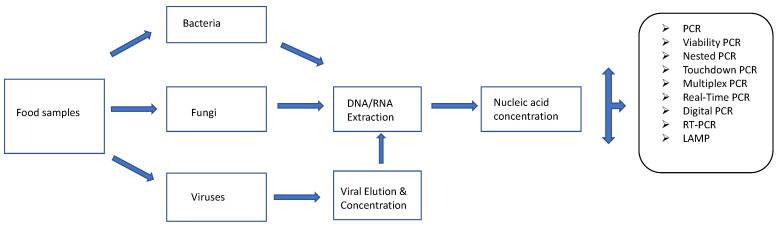
Nucleic acid-based methods for detecting and identifying foodborne pathogens. Adapted from [92].

**Figure 2 microorganisms-11-01111-f002:**
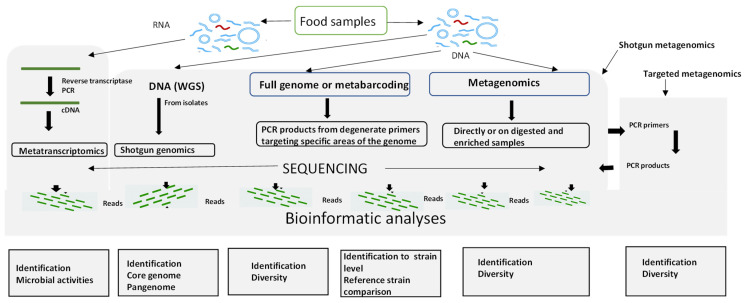
Next-Generation Sequencing approach for detecting and identifying foodborne pathogens. Adapted from [32,133].

**Table 1 microorganisms-11-01111-t001:** Methods for foodborne pathogen detection.

Method	Target Groups	Principle	Advantages	Limitations
Culture Based Methods	Bacteria, fungi, and viruses	Use of culture media containing the growth requirements of target microorganisms	High success rate for culturable isolates Reliable and cost-effective Can be used to target specific microbial groups & differentiate different groups Microbial cultures can be used for other applications or downstream processes	Misses large amounts of microbial groups that cannot be cultured (low sensitivity Slow-growing cultures are disadvantaged Slow turnover and laborious processes Not suitable for the rapid detection of microorganisms
Immunological assays (ELISA and LFD)	Bacteria, fungi, and viruses	The affinity between microbial antigens and monoclonal or polyclonal antibodies is exploited for the detection of microorganisms	Easy to carry out as the process can be automated improving efficiency and making it less labour intensive Large number of samples can be processed at once Can be highly specific, Toxins can be detected	False positives can be generated Limited microbial coverage Require the use of trained personnel antibodies/antigens must be labelled
Conventional PCR (e.g., nested, touchdown, multiplex, etc.)	Bacteria, fungi, and viruses	Based on the use of primers targeting specific regions of microbial DNA. Target when present is exponentially amplified	Highly sensitive and specific Reliable and widely used Can be automated Can be used to detect multiple microbial groups in a single reaction	Primer design is very important Difficult to use to distinguish between viable and non-viable cells Sensitive to contamination and this may lead to false positives Sensitive to inhibitors which may lead to false negatives
Viability PCR/qPCR	Bacteria	Special dyes used in the test render non-viable bacterial DNA non-amplifiable	Dead and viable bacterial cells can be easily differentiated Viable cells can be quantified Faster than culture-based approaches	False positives can result Use of mRNA more reliable than DNA Not as widely used as other methods
Reverse Transcriptase (RT) PCR/qPCR	Bacteria, fungi, and viruses	mRNA transcripts level declines quickly after cell death. Detected mRNA should be from viable cells and is therefore targeted (RT-PCR). Transcripts are amplified and quantified using dyes or probes	Relatively quicker than culture-based approaches Reliable and widely used Highly sensitive and specific Quantification of cells (absolute number or gene copy numbers) can be achieved	False positives can occur as not all mRNAs are short-lived Requires the use of trained personnel Can be expensive to run Not suited for use when samples yield < 200 bp products Sensitive to contaminants and inhibitors
Next Generation Sequencing Approaches	Bacteria, fungi, and viruses	Whole genome or markers such as ITS and 16S rRNA are targeted and sequenced (random shotgun sequencing, gene/marker specific sequencing, etc.)	Reliable and accurate results and also now more widely used Suitable for detection/identification of all groups of microorganisms (rare, fastidious & unculturable groups Predictable turn-around times Large number of samples can be analyzed Generates a huge amount of useful data Continuous improvement of technologies and platforms	Expensive to set up and run compared to other methods Requires trained personnel to run and analyze the results Result are as good as the reference database

Adapted from [3,6,15,16,17,37,38,39,40].

## Data Availability

No new data were created or analyzed in this study. Data sharing is not applicable to this article.

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
