# Peer review of "A Review of Modern Methods for the Detection of Foodborne Pathogens"

_microorganisms, 2023, doi:10.3390/microorganisms11051111_

Round 1

Reviewer 1 Report

This is an excellent review and should be considered for publication.  I suggest adding 2-3 more tables and can provide a summary related to current methods, advantages, and disadv.  

it should be a good idea to add one paragraph related to sampling for microbial analysis and standard methods for testing.  AOAC or ASM. 

the conclusion is very limited and does not describe some limitations and future needs.  overall excellent job. thank you for the effort. SAI

Reviewer 2 Report

The work is interesting and addresses an important public health topic. The work is well organized and described in an understandable way. The work is scientifically valid and not misleading. The bibliography is adeguate and relevant.

Author Response

I am very grateful for the reviewers' comments.

Reviewer 3 Report

The review is well conducted and I have proposed some questions that I think should be addressed.

- Delete the examples in lines 9, 15, 17 and 18 because I think it is more important improving the conclusion in the abstract, explaining the impact of your review and the main discoveries.

-Line48 delete (nisin).

-Conclusion and future directions: I believe that future trends and directions should be included. Authors can discuss, for example, the latest technology and alternatives to preservatives.
-References provided should be well formatted and I suggest avoiding articles older than 10 years whenever possible.
